# Polygon Simplification for the Efficient Approximate Analytics of Georeferenced Big Data

**DOI:** 10.3390/s23198178

**Published:** 2023-09-29

**Authors:** Isam Mashhour Al Jawarneh, Luca Foschini, Paolo Bellavista

**Affiliations:** 1Department of Computer Science, University of Sharjah, Sharjah P.O. Box 27272, United Arab Emirates; ijawarneh@sharjah.ac.ae; 2Dipartimento di Informatica—Scienza e Ingegneria, University of Bologna, Viale Risorgimento 2, 40136 Bologna, Italy; luca.foschini@unibo.it

**Keywords:** approximate query processing, Spark Streaming, stratified sampling, spatial sampling, Douglas–Peucker, line simplification, geospatial generalization, mobility data, low-cost air quality sensors data, pollution data

## Abstract

The unprecedented availability of sensor networks and GPS-enabled devices has caused the accumulation of voluminous georeferenced data streams. These data streams offer an opportunity to derive valuable insights and facilitate decision making for urban planning. However, processing and managing such data is challenging, given the size and multidimensionality of these data. Therefore, there is a growing interest in spatial approximate query processing depending on stratified-like sampling methods. However, in these solutions, as the number of strata increases, response time grows, thus counteracting the benefits of sampling. In this paper, we originally show the design and realization of a novel online geospatial approximate processing solution called GeoRAP. GeoRAP employs a front-stage filter based on the Ramer–Douglas–Peucker line simplification algorithm to reduce the size of study area coverage; thereafter, it employs a spatial stratified-like sampling method that minimizes the number of strata, thus increasing throughput and minimizing response time, while keeping the accuracy loss in check. Our method is applicable for various online and batch geospatial processing workloads, including complex geo-statistics, aggregation queries, and the generation of region-based aggregate geo-maps such as choropleth maps and heatmaps. We have extensively tested the performance of our prototyped solution with real-world big spatial data, and this paper shows that GeoRAP can outperform state-of-the-art baselines by an order of magnitude in terms of throughput while statistically obtaining results with good accuracy.

## 1. Introduction

The unprecedented overdependence on the ubiquitous Internet of Things (IoT) in all aspects of our lives has caused the accumulation of massive amounts of multidimensional and heterogeneous data that are mostly georeferenced [1,2,3,4]. To extract insightful knowledge from such an abundance of data, the fast-track adoption of a breed of big data processing frameworks such as Apache Spark has occurred.

In particular, with most data from IoT arriving in streams, unprecedented efforts in the relevant literature have focused on designing and supporting the promotion of a constellation of tools known as data stream processing systems (SPSs) [5]. Those systems typically search for a good trade-off between two indispensable QoS requirements specified in service-level agreements (SLAs): low latency and high accuracy. As those requirements are normally contradicting, where improving one of them leads naturally to the deterioration of the other, a constellation of different SPSs, characterized by different trade-offs and based on approximate computation, has recently emerged [3,4,6,7]. Approximate computing is driven by the fact that users are typically satisfied with approximate results and willing to give up tiny error-bounded losses in accuracy in exchange for a lower latency [4,6]. There are numerous methods applied in approximate computing. However, they are mostly based on either selecting subsets of the arriving data streams or discarding (also known as shedding) extra data loads that may otherwise introduce additional latencies. Sampling is, by and large, one of the most widely adopted methods for such purposes [3].

Embracing randomness has been the defining factor for most sampling methods that have been designed in the relevant literature so far. This, in part, is attributed to the fact that application scenarios were increasingly known to generate data that exhibited normal distributions [3]. However, things have changed dramatically with the emergence of smart environments (e.g., smart cities) and big data scenarios. Currently, those scenarios are generating big data characterized by high skewness and thus are non-uniformly distributed. In short, this means that the dependence on random-based sampling methods is not going to perform well when it comes to approximating geo-statistics (e.g., the average speed of a vehicle moving in a specific trajectory in a city).

Online sampling that is aware of the geographical characteristics of big data has been demonstrated to be computationally expensive and could cause SPSs to halt at high spikes in data arrival rates [6]. This problem is exacerbated by the fact that well-performing spatial sampling algorithms have to be based on advanced data structures as opposed to simple random sampling. In fact, they are mostly based on stratification, which divides the data into groups corresponding to the real geometrical areas where they have been collected in order to be able to draw roughly equal amounts of data objects from each area independently.

In this paper, we propose a new method called Geospatial Reduced Approximate Processing (GeoRAP). It is a novel geospatial online sampling solution that can efficiently generate highly representative spatial samples within the premises of stringent time-based QoS constraints (i.e., high response time and lower latency). GeoRAP employs a geometric line simplification (i.e., generalization) algorithm to simplify the polylines of the covering polygons (i.e., the border lines of the administrative divisions from where the geospatial vector data are usually withdrawn). Our solution adopts the Douglas–Peucker (hereafter DP for short) algorithm (also known as Ramer–Douglas–Peucker) [8]. We first pass the geometrical administrative divisions (often in shapefile and GeoJSON formats) to a frontline filter that applies a line simplification algorithm to simplify the border lines (typically by minimizing the number of vertices of polygons); then, we pass the resulting new covering polygons to a geohash encoder that generates the list of geohashes covering each polygon. Geohash encoding is a dimensionality reduction approach that transforms the 2-D parametrized long/latitude points into a corresponding string-based representation that represents a Minimum Bounding Rectangle covering the original point. Thereafter, upon receiving a stream within each time window interval, our method selects a fair amount of the arriving geospatial tuples from each polygon independently. The crux of GeoRAP is that it minimizes the size of strata for each polygon (each geohash is a stratum in this case). This is a direct consequence of simplifying the borderlines, thus minimizing the number of vertices.

Various vector line generalization algorithms are discussed in the literature and divided into five categories, as in [9]. Examples include the Reumann–Witkam routine [10], the sleeve-fitting polyline simplification algorithm [11], the Lang simplification algorithm [12], the Douglas–Peucker algorithm [8], and an algorithm by Visvalingam and Whyatt [13]. The difference between the global routines (e.g., DP) and the other categories is that the former takes the entire line into consideration while processing, whereas other categories take segments of the entire line gradually from a start point to an end point of the original line. DP is the preferred algorithm in the state-of-the-art, for it is the most accurate in preserving the shape of the original line, as discussed in [9]. It is for this reason that we adopt the DP algorithm as part of our GeoRAP system in this paper.

In short, we believe that this paper can provide the readers with the following valuable and original contributions. We designed a novel online geospatial sampling solution to facilitate on-the-fly geo-statistics (e.g., estimating the average speed of all vehicles moving during a time range in a metropolitan city). We designed, implemented, and validated a sampling method based on an industry-standard SPS, i.e., Spark Streaming [5]. Our implemented prototype shows that it is capable of lowering the latency while keeping the accuracy loss in check and within acceptable thresholds. To the best of our knowledge, no other implemented system solution from the literature achieves these goals.

The remainder of the paper is structured as follows. We first discuss relevant studies from the recent state-of-the-art in Section 2. In Section 3, we introduce appropriate theoretical concepts related to our work and provide the readers with the needed background. We then describe the primary design and implementation choices we made in our development and implementation work in Section 4. In what follows, we discuss our experimental validation work and the associated performance results in Section 5. Conclusive remarks and recommendations about future research frontiers end the paper in Section 6.

## 2. Literature Review

### 2.1. Spatial Approximate Query Processing

Unprecedently, massive amounts of big, geotagged data streams are hitting database management systems (DBMSs) continuously, with fluctuations in such a way that in peak hours, data size typically overwhelms those systems and far exceeds their computational capacity [3,6,7]. Seeking to derive insights from such data [6], DBMSs need to manage and store such data efficiently. In such cases where size is overwhelming, systems resort to either elasticity or adaptivity [14]. In the former, systems overprovision or deprovision resources depending on the fluctuation in data arrival rates, whereas in the latter, systems resort to applying approximate query processing (AQP). The disadvantage of elasticity is that it keeps the online DBMS busy with configurations while those resources can be better allocated to data stream processing instead.

The typical form of adaptivity is represented by AQP. This entails reducing the size of the arriving data in such a way that enables the system to operate gracefully in response to the fluctuations in data arrival rates. The most adopted method for AQP in the literature is sampling, which works by selecting a representative subset of the arrived data in a way that guarantees bounded error numbers in exchange for significant gains in processing times and reduced latencies. AQP is highly sought after in the state-of-the-art simply because the tiny loss in accuracy in exchange for a significant gain in speedup is desirable and acceptable by decision makers in smart city scenarios [4].

Authors in [7] proposed an AQP system for online data stream processing. Their system is composed of three main components: data learning, dynamic sampling strategy, and error control. Their sampling strategy is based on stratification and binary-trees division. They split arriving data streams into tree-based nodes (nodes are analogous to partitions in data distributed parallel processing terms). The other component is a loop-feedback-based error controller to compute the sample sizes adaptively for subsequent processing cycles. The shortcoming of this framework includes the fact the stratification methodology is expensive as it is tree-based. Also, the system is unaware of spatial characteristics and is not reliable for spatial online data processing.

In the same vein, SnappyData [15] has been designed to operate over Apache Spark [16] and incorporates AQP for data stream processing and analytics. It features data structure modules that balance latency and accuracy QoS SLA targets. However, it is not optimized for spatial AQP and data analytics. It does not host a spatial-aware sampler.

Authors in [17] have designed Taster, which encapsulates a few components, including a collector that employs synopses reduction techniques as front stages to select a subset of data. In addition, it hosts a tuner and cost planner, which is responsible for producing on-the-fly approximate running plans. Plans are then run against a synopsis (e.g., samples or sketches) that either persists in a data warehouse or is poured from an online data stream. Plans should be able to meet the accuracy/latency targets from an SLA. Those plans are passed to a tuner, which is then responsible for selecting the plan that optimizes the performance. The system then utilizes the selected plan to capture the synopsis (persisted in disk or from online data streams). The system then computes approximate results from those synopses to answer a query posed by the user. This system, however, is not designed with spatial data processing in mind; it also regularly stores synopsis in disks, which adds extra overhead.

Stock versions of those AQP systems do not include over-the-counter support for geospatial data processing, which leaves handling those logistics to the presentation layer, thus overwhelming the shoulders of front-end developers and distracting their focus orientation away from the main task of developing dependable spatial data stream analytics. This is so because geospatial data are parametrized and represented in the form of pairs of coordinates (typically longitudes and latitudes). Having said that, spatial data lose their characteristics while moving and floating around in a network, and bringing them back to their multidimensional shape is a computationally expensive task that normally involves costly spatial join processing [18,19]. In conclusion, for a data stream processing system to be able to work with big geospatial data at scale with QoS guarantees, it must feature by-design products that incorporate support for geospatial data processing and be aware of other spatial characteristics intrinsically. It also should support interactive and proactive mechanisms to respond to sudden spikes in data arrival rates in such a manner that guarantees, to a good extent, the stability of the system. It should be well-noted, however, that online spatial sampling is challenging basically because of the multidimensionality of data. Stock versions of state-of-the-art data stream processing engines do not currently offer spatial awareness and AQP with QoS guarantees together [3,6,19,20].

Within the same consortium, ApproxHadoop [21] features a two-level sampling mechanism atop Apache Hadoop, which drops tuples to reduce data size in order to reduce I/O overhead. Nevertheless, it relies on an offline profiling method for reconfiguring the latency/accuracy estimates and, therefore, is hardly applied for data streaming workloads.

Other big data systems employ sampling for spatial data partitioning in the Cloud. For example, Simba [22] and SpatialHadoop [23]. However, they do not work for dynamic spatial data stream processing scenarios, where data arrival rates fluctuate periodically.

Only a few systems in the recent state-of-the-art offer such capabilities. Perhaps most important is a system we designed in a previous work, which we termed ApproxSSPS [4]. It basically hosts two state-of-the-art spatial-aware online sampling methods (SAOS [6] and ex-SAOS [3]) that we have designed previously for selecting representative spatial samples on the fly with QoS guarantees. It also features a controller that senses interactively the fluctuation in data rates and responds by selecting the percentages of the samples to be withdrawn from the online stream. However, the system employs sampling on the original polygons representing the study area, which could reduce the efficiency in cases of very big data streams.

Also, EXPLORA [24] has been designed to support AQP and geo-visualization of spatiotemporal data at scale with QoS guarantees. It hosts a sampling method that selects online representative samples of arriving data, which aims to speed up processing while keeping accuracy in check. However, it does not host a latency-aware controller that pulses the fluctuation in data arrival rates and reacts proactively.

Also, spatial-aware approximate query processing (SAQP) is employed for integrating calibrated granular heterogeneous spatially tagged pollution and mobility data for joint analytics at scale with QoS guarantees, as in [25]. Authors have designed EMDI (Environmental Mobility Data Integrator) for integrating mobility data with environmental data, e.g., pollution, climatological, and meteorological data, for complex unified analytics with QoS guarantees. It features a sampler that samples data from both worlds for unified AQP analytics.

However, those systems do not take advantage of the line simplification algorithms. As stipulated in Tobler’s first law of geography, objects that are closer in real geometries are more related. Taking advantage of this fact has been widely proven in the literature [3,4,6,26,27]. Having said so, most analytics do not consider spatial objects floating on the outskirts of cities, and most analytics are focused inside cities and livable areas. Furthermore, line simplification algorithms are indispensable for efficient, timely analytics of big georeferenced data with QoS guarantees, such as reducing latency while keeping accuracy in check.

### 2.2. Line Generalization Algorithms

Vector line generalization algorithms can be divided into several categories, as discussed in [9]. Those are independent point algorithms (such as the nth point routine and the routine of random selection of points), local processing routines (such as the routine of distance between points and the perpendicular distance routine), unconstrained extended local processing (such as the Reumann–Witkam routine [10] and the sleeve-fitting polyline simplification algorithm [11]), constrained extended local processing (such as the Opheim simplification algorithm and Lang simplification algorithm [12]), and global routines (such as Douglas–Peucker algorithm [8] and an algorithm by Visvalingam and Whyatt [13]). For example, the independent point algorithms work by grouping points on a line into independent sets of consecutive points and then retaining random points. Local processing routines depend on the idea of eliminating points within a radial or perpendicular distance that is less than a user-supplied threshold (tolerance bandwidth) while retaining points within a distance greater than that of the tolerance threshold. From the family of unconstrained extended local processing, the Reumann–Witkam routine [10] works by first passing a strip (rectangular) that shifts stepwise through all segments of an original line, where the start and end points within the strip are retained while in-between points are eliminated. The sleeve-fitting polyline simplification algorithm works in a way similar to that of the Reumann–Witkam routine, where the strip is known as a sleeve that is passed through the original segments of the original line. The Opheim simplification algorithm from the category of constrained extended local processing algorithms works in a way that is similar to that of unconstrained extended local processing routines, in addition to minimum and maximum additional constraints applied within a search region (similar to strips and sleeves), where points within the minimum tolerance or within the search region bounded by the maximum tolerance are removed, while the other points of the original line are retained. Lang simplification algorithm works in a similar way as compared to the Opheim simplification algorithm utilizing search regions, in addition to a distance of intermediate points to a segment connecting start and end points in the region, which should not exceed a user-supplied tolerance for in-between points to be retained. A comprehensive review of those methods and algorithms is beyond the scope of this paper. In this paper, instead, we concentrate on methods that apply those algorithms for various vector geospatial data processing workloads.

Global routines, e.g., DP, take the entire line into consideration while processing, contrary to what appears in the other four categories, which consider segments of the entire line stepwise from the start point to the end point. Visvalingam–Whyatt from this category works on the basis of eliminating the original point with the minimum effective area (triangular area formed by any point and its direct in-between neighboring points). The DP algorithm performs favorably for tiny weeding (minor simplification), while the Visvalingam–Whyatt algorithm wins for eliminating entire shapes (e.g., caricatural generalization) [9].

Extensive experiments conducted by [9] show that the Douglas–Peucker algorithm produces the most accurate generalization in terms of measures such as mean distance and mean dissimilarity (which means it has the best performance in the language of shape distortion). For this reason, we adopt the DP algorithm as part of our GeoRAP system in this paper. The Douglas–Peucker algorithm is discussed in detail in Section 3.4.

### 2.3. Applications of Line Simplification in Approximate Geospatial Analysis

Line generalization has been widely adopted in several aspects related to complex geospatial analysis, including trajectory compression. For example, a recent work by [28] proposed to parallelize several trajectory compression algorithms atop Apache Spark, including an algorithm that is based on Douglas–Peucker. So, it has been applied to trajectories formed from GPS points so that trajectories are lines formed by connecting points, and then parts of those lines are cut, and their corresponding points are discarded accordingly. The process does not involve reducing a polygon; thus, no spatial join is involved.

In the same vein, ref. [29] have designed a novel geospatial-aware big data analytics framework that features trajectory compression using DP algorithm line simplification as an integral part of its operation. The same applies to a work by [30], where authors proposed an Enhanced Douglas–Peucker (EDP) algorithm which applies a group of enhanced spatial–temporal constraints (ESTC) while simplifying trajectory data, aiming basically to preserve few spatial–temporal features while applying the line simplification for compressing trajectories.

Within the same consortium, ref. [31] designed a novel method to unleash hotspot routes in mobility data by employing massive taxi trajectory data. To reduce the burden on storage and processing resources, they employed the DP algorithm as a quick-and-approximate filter in the front stage of their system to reduce the number of trajectory points accepted for clustering downstream in their system.

Also, variations of the plain application of the DP algorithm are available. For example, ref. [32] proposed a velocity-preserving trajectory DP-based simplification algorithm, which divides the source trajectory into sub-trajectories based on their average velocity such that the velocity in every sub-trajectory differs largely from the others. This way, each sub-trajectory is assigned a different threshold based on those figures; thereafter, each sub-trajectory is simplified using the plain DP algorithm independently, then all simplified sub-trajectories will be merged into a single simplified trajectory.

In addition, ref. [33] designed an adaptive DP-based algorithm with automatic thresholding for AIS-based vessel trajectory compression. Also, another DP-based algorithm has been designed by [34] for compressing automatic identification system (AIS) ocean massive datasets. Also, in oceanography, refs. [35,36,37] designed a DP-based algorithm for the compression of AIS ocean ship data.

Another line of applications of line simplification algorithms for scale-aware geospatial data analytics includes big georeferenced data visualization. In this direction, ref. [38] applied a DP-based algorithm for reducing the georeferenced data to be visualized at scale with QoS guarantees. Authors in [39] propose a novel algorithm that they term locally adaptive line simplification (LOCALIS) for GPU-based geographic polyline data visualization at scale with quality guarantees. Similar work appears in [40], where a method for Level-of-Details (LoD) visualization for polygonal data was designed, which incorporates a polygon simplification method that is based on the DP algorithm. Ref. [41] applies the DP algorithm for generating trajectory data for thematic heatmaps at the city scale for tourist activity analysis. Similarly, authors in [42] have designed an approach for the efficient generation of heatmaps using methods based on the DP algorithm. Also, ref. [43] designed RectMap by combining DP-based simplification with a gridding algorithm to generate simplified versions of plain reference maps.

The picture that emerges from the recent state-of-the-art is the following: systems are mostly run on beefed-up servers and not deployed in parallel. Also, they are not applied with stratified-like spatial sampling, and they are mostly applied for simplifying polylines, not polygons. Simplifying polygons has a great impact on the processing and qualified analytics of big geospatial data streams for several reasons, including the fact that georeferenced sensor data are parametrized and typically need to be joined with polygons for insightful analytics, which adds extra I/O and computational overheads. Also, sending those polygons to the Edge node in Edge–Cloud deployments that are designed for spatial data analytics means adding extra layers of network overhead that slow down the performance overall. Also, federated learning is a line of machine learning (ML) that parallelizes ML algorithms so that they run on Edge devices near the data, so being able to join the spatial data quickly on those devices with polygons data can provide significant speedups.

## 3. Theoretical Background

### 3.1. Spatial Sampling in Dynamic Scenarios

Most smart city analytics involve queries that seek a correlation between georeferenced entities based on their geographical proximity. For example, a query could ask for “taxi pickup density for each neighborhood in NYC across time”. The density itself is what concerns the analyst, not the exact number of pickups in every neighborhood. That said, a sample from each neighborhood that reveals the differences in densities between all neighborhoods is typically acceptable [3]. Since huge amounts of multidimensional georeferenced data arrive from IoT devices, cases where data stream processing engines are not able to cope with the data arrival rate and fluctuation are not unheard of [4]. For those peak periods, analysts normally are willing to give up tiny losses in the accuracy of trade for significant gains in time-based QoS goals such as response time [6]. By far, spatial sampling remains one of the most attractive solutions for reducing the amount of data streams to be analyzed by SPSs downstream. This is attributable to the fact that geo-statistics that are based on approximation of the results with rigorous error bounds are acceptable and sufficient for smart city analytics [3,4,6].

### 3.2. Representative On-the-Fly Geospatial Sampling

In the literature, there are two main approaches that deal with the fluctuation in the arrival rates of big data streams. Stream processing systems depend on either the dynamic assignment of extra computing resources (also known as overprovisioning resources) or data reduction approaches (e.g., sampling and sketches), thus applying approximate query processing (AQP) paradigms and trading off tiny accuracy losses for a significant gain in response times [6]. Both solutions are attractive for coping with changes in data arrival characteristics. However, the former is less desirable as it tends to waste resources whenever the arrival rate slows down, which could mean the underutilization of scarce computing resources that could otherwise be utilized elsewhere. The latter is typically based on simple random sampling (SRS) [44]. Tobler’s first law of geography dictates that everything is related to everything else; however, things that are close are more related than things that are far apart [45]. Since in geospatial analytics, we are mostly concerned with unveiling correlation facts regarding the near geospatial things (i.e., geospatial entities), a sampling design that considers the geospatial distribution of spatial entities in real geometries is, therefore, more desirable, even if for no other reason but to generate approximate results for geo-statistics with statistically plausible rigorous error bounds. With that in mind, it is, thus, desirable and attractive to be able to select geospatial representative samples from the arriving data streams. By representativeness, we mean that the minuscule selected from the arriving data leads to very accurate approximate results for most of the geo-stats with rigorous error bounds that are below defined thresholds. This kind of sampling design is typically known as *geospatial representative sampling* [3,6].

### 3.3. Problem Formulation

In this section, we provide a few formal definitions in an order that works as a proxy to help comprehend the mechanism by which the main constituting components of our novel system, GeoRAP, operate.

**Definition 1 (Geospatial data).** *A spatial dataset consists of several georeferenced data tuples on the form (x, y, [values]), where x and y are geo-coordinates (either geographic or projected coordinate system), such that *$D=[\left(x_{1},y_{1,}{values}_{1}\right),\left(x_{2},y_{2},{values}_{2}\right),\ldots \ldots ..,\left(x_{n},y_{n},{values}_{n}\right),\;\left|D\right|=n$* is the number of tuples in the dataset. Geospatial data can be encoded with a geographic encoding (such as geohash), thus generating the following list*:
$$D=[\left(x_{1},y_{1,}{values}_{1},geo1\right),\left(x_{2},y_{2},{values}_{2},geo2\right),\ldots \ldots ..,\left(x_{n},y_{n},{values}_{n},geon\right),\left|D\right|=n$$

**Definition 2 (Geospatial data distribution).** *The embedding space containing a geospatial dataset is overlayed with a grid of regularly shaped (equal or non-equal sized) tiles (rectangles, for example, geohash covering). Geo-coordinates and probability distributions of tiles combined form a histogram *$p=\left[p_{1},\;p_{2},\ldots \ldots .,p_{n}\right],$ *where the sum of all probabilities equals 1, *$\sum_{i=1}^{n}p_{i}=1$.

**Definition 3.** *(Reduced) geo-cover. A geohash cover is the list of all geohash values that cover the polygons of a polygon file representing the study area and equals *$cover=[g_{1},\;g_{2},\;\ldots ..,\;g_{n}]$. *Reduced geohash cover is a geohash cover that overlays the polygon file generated by Douglas. If ‘number of PTR’ = m, then reduced geohash cover, *$reducedCover=\{g_{1},\;g_{2},\;\ldots ..,\;g_{m}\}$*, such that *reducedCover⊂ cover*, and *m ≤n.

**Definition 4.** *Geospatial sample. Suppose we have a geospatial data population on the form *$P=\{p_{1},\;p_{2},\;\ldots ..,\;p_{mn}\}$*, a geospatial sample is a subset of a population geospatial data such that *$S\subset \;P=\{p_{1},\;p_{2},\;\ldots ..,\;p_{m}\}$*, where m is the size of the sample such that *m ≤n. *A geospatial sample selected based on the reduced geohash cover as per Definition 3 is as follows; *$S_{reduced}\subset \;P=\{p_{1},\;p_{2},\;\ldots ..,\;p_{k}\}$*, such that *k≤m ≤n.

**Definition 5.** *RDP-based geospatial Top-N query. Given a geohash encoded dataset such as the one defined in Definition 1 and a reduced covering geohash such as the one defined in Definition 3, based on a sample drawn based on the reduced geohash cover from Definition 4, a Top-N query is defined as it follows, result = *${First}^{N}(sort\left(\overset{m}{\underset{i=1}{⋃}}\left[{polygon}_{i},\;{count}_{i}\right]\right))$*, where m is the number of polygons, N is the number of Top-N polygons specified in the Top-N query, *${First}^{N}$* is the top N entries after sorting*.

### 3.4. Geometric Generalization

Geometric generalization simplifies map content for the preservation of readability and comprehensibility, thus creating downscaled maps. One of the most adopted methods is the simplification of polylines, which results in a reduction of the geospatial points with no change of coordinates.

In cartography design, map generalization is the process of deriving small-scale maps from large-scale counterparts by applying well-defined changes that are performed either manually by a cartographer or through computer algorithms. The basic function of generalization is to abstract geospatial data that are represented with high levels of detail into a lower level of detail representation that can seamlessly be rendered on maps. Maps are considered well-generalized if they succeed in emphasizing the most relevant elements of a map while representing the real world recognizably well.

There are several cartographic methods that can be employed to modify the amount of geospatial data on maps, including selection, simplification, smoothing, merging, aggregation, and typification. Simplification is the process of removing vertices from lines and polygonal area boundaries [46]. Additionally, line simplification algorithms are normally categorized into two groups: filtering and other algorithms. While filtering algorithms retain part of the original vertices in boundaries and lines, other algorithms (e.g., smoothing) fit a set of new points over the original points.

In that sense, line simplification typically removes fewer contributing vertices (unwanted details), thus minimizing large-scale detailed geospatial data in order to render it on constrained screens and reduced scales. Various approaches have been proposed, where, despite discrepancies, they mainly involve searching vertices of lines, discarding those that have little contribution to the general shape of the line. The Ramer–Douglas–Peucker algorithm is, by far, the most common approach for line simplification that we have found in the literature [8].

The Douglas–Peucker (also known as Ramer–Douglas–Peucker, RDP hereafter for short) algorithm [8] is a geometric line generalization filtering algorithm that operates by reducing the number of vertices of polygons representing a geographical study area. An overarching trait is that it preserves the polygonal shape characteristics within certain limits. It achieves statistically plausible results for natural geometries (e.g., forests, boroughs). Figure 1 shows the graph of GeoJSON of the Chinese city of Shenzhen before and after applying line simplification using the RDP algorithm. It is well-noted that the shape is not significantly distorted. However, the number of vertices is reduced significantly. A configurable parameter in the algorithm is the “percentage of removable points to retain” (we refer to this as “number of PTR” hereafter for brevity), a number that takes a value between 0 and 100, dictating how aggressively to simplify the geospatial polygons. So, a value that is equal to 1% is a very aggressive simplification, saving a huge amount of hard disk space. The product of the algorithm is a list of simplified lines that remain within a specified distance of the original line. It is known to be effective in thinning dense vertices of polygons; however, it could form spikes at high simplification levels.

Figure 2 shows an example application of the Ramer–Douglas–Peucker line simplification algorithm to the administrative polygons representing NYC in the USA.

Figure 3 shows the mechanism by which the RDP algorithm operates. Given a polyline L and a threshold value *alpha*, the goal is to construct the line between the start (termed anchor) and end points (termed floater) of L. The algorithm starts by finding the point in the polyline (an intervening point) that is the furthest from the line segment. If the distance between that point and the line segment is greater than the threshold, then the point is significant; the algorithm then constructs a line segment between the start point and the significant point and another line segment between the significant point and the end point. The same procedure is then recursively repeated for both line segments until the threshold criteria are satisfied. For any test, if no significant point is found, the algorithm simply removes all the points between the start and end parts of each line segment, as this segment is considered adequate for the simplified representation of the line.

A more recent line generalization algorithm is known as Visvalingam–Whyatt [13]. It works by iteratively removing the so-called “least important point from a polyline”. It measures the importance of points by employing a metric that is based on the geometry of a triangle that is formed by every non-end-point vertex and two neighboring vertices. In other terms, the Visvalingam algorithm employs the so-called “effective area” metric points forming smaller-area triangles, which are then removed first. This results in removing smaller-angle vertices. In other words, it employs characteristics of triangles within polygon representations to decide the vertices to remove.

The importance of line simplification stems from the fact that in smart city dynamic application scenarios, we require interactive visualization of fast-arriving georeferenced data streams, where line simplification reduces display times. This is our intention because the data we are sampling is going to be used for interactive geo-visualization.

For example, in region-based aggregate geo-maps such as choropleth thematic mapping, line simplification can be employed to diminish administrative boundaries from the polygons representing a city while emphasizing foreground information in the geospatial distribution (e.g., count of vehicles in each district of a city).

## 4. Representative Geo-Sampling for Dynamic Application Scenarios: An Overview of the GeoRAP Solution

In this section, we showcase the design and implementation of our novel system, Geospatial Reduced Approximate Processing (GeoRAP for short), for efficient processing of big georeferenced data at scale, with QoS guarantees. We start by explaining a case scenario from smart cities that motivates the need for our system, in addition to discussing a plain baseline with which we compare our novel system.

### 4.1. Case Scenario and Baseline Systems

Urban planning plays an integral role in improving the quality of life of city inhabitants from various perspectives. Decision makers normally require high-level views of factors that are typically in interplay in shaping the dynamicity of metropolitan cities. For example, they require the geo-visualization of georeferenced big amounts of heterogeneous data streams in real-time so that they monitor the progression of various phenomena, such as the trends at which densities of vehicle’s mobility in specific points of interest within the city across time are changing. Those views normally take the form of real-time heatmaps and interactive dashboards. However, since the geotagged arriving data streams in such smart city scenarios are normally huge and overwhelming, cases where geo-map generation takes a few minutes or even hours are not unheard of [3]. This stands as a clear prohibitive obstacle in the way of timely decision making. Furthermore, display devices are normally resource-constrained and, thus, are unable to draw pixels in correct ways that reveal the real geometrical facts (e.g., the density of moving vehicles). Consider a realistic example where a municipality administration is requesting the generation of interactive on-the-fly heatmaps (or similarly density maps) that show the dynamicity of moving geospatial entities (e.g., people, bikes, cars, e-scooters, etc.) within the city to reveal the correlations between those densities and traffic jams across time. At specific times across a weekday, those moving objects are normally clumped into a few patches (e.g., school areas, city center, etc.), which potentially will result in cluttered geo-maps. Stated another way, even if the SPS could ingest all the arriving data, it should not do so as those cannot be drawn simultaneously on maps because of the map space constraints. A natural solution in this case scenario would be the ability to choose a well-representative, well-spread-out geospatial sample that fairly selects roughly the same amount of spatial moving objects from each administrative division (i.e., district, borough) in the city independently, which is known to preserve spatial characteristics (e.g., density distribution), thus yielding more accurate approximate geo-maps with rigorous error bounds [3,4,6].

Our methods are not directly comparable with those of the recent state-of-the-art that are discussed in Section 2. To the best of our knowledge, we have not found any system from the recent literature that provides system features that could be comparable to those introduced in our system, GeoRAP. Having said that, our baseline is based on a plain implementation of the algorithms without the application of line simplification algorithms. Specifically, to provide the rationale for the introduction of our new system in this paper, we are highlighting a conventional baseline system for selecting geospatially representative samples in smart cities. Specifically, we are comparing the new system GeoRAP for representative geospatial reduced online sampling with a plain stratified-based sampling baseline [6]. The baseline is a fast-memory online spatial sampling method that is intrinsically embedded with Spark Structured Streaming and synergistically supports its capabilities for the applications of approximate analytics of huge amounts of geotagged big data streams.

The baseline, in its essence, is a stratification-based method. In the plain implementation of the method, the same percentage of points for each geohash during each time interval (known as batch interval in online stream processing parlance) is selected. Geohashes can be heuristically thought of as grid squares resulting from the division of flattened planar geometry (the survey area). The plain method simply works by first calculating the covering geohashes for each administrative part of a city (normally known as neighborhoods), thereafter selecting the same percentage from each group points having the same geohash arriving during a batch interval, which means that we approximately choose a fair number of points from each neighborhood.

However, referring to Tobler’s first law of geography [45], geospatial analysts and data scientists are more interested in city areas where most geospatial objects are clumped into few patches (e.g., school areas), thus causing problems to the SPS while they try to ingest the huge amounts of naturally correlated geo-data at rush hours. City bordering areas are normally significantly less congested thanks to the highways and autoroutes. Since there is less data arriving from those areas as compared to within-city areas, it makes sense to simplify the city map on bordering areas, removing some vertices from the maps. A spatial stratified-like sampling design operates by selecting roughly the same percentages of geo-entities from each city administrative division. However, because of the natural design of stratified spatial sampling that is based on geohash as a stratification method, depending on the precision of the geohash, the number of stratum (i.e., strata) can grow significantly, causing unnecessary bottlenecks. This is mainly avoidable if we consider that we can discard geohashes resulting from bordering areas of a city. This is exactly the main essence by which our novel method, GeoRAP, in this paper operates.

### 4.2. GeoRAP Design and Operation

We have designed GeoRAP so that it operates in two modes. The first mode is a batch (i.e., static) mode, where we have a disk-resident georeferenced big dataset, while the second mode is a dynamic one, where we have data that indefinitely pour downstream in a pipeline workflow toward an ingestion layer that ingests the data and feeds them to our system regularly on a timely basis. The high-level architecture of our GeoRAP system is shown in the context diagram of Figure 4.

The starting point of operation occurs at the front stage, where GeoRAP hosts a *borderline simplifier* (currently supporting RDP algorithm) component. This component receives a polygon file (in the form of GeoJson or shapefile) representing the administrative polygons (i.e., regions, districts, neighborhoods, boroughs, etc.) of the study area (e.g., city). It then applies the RDP algorithm to simplify the polygons, thus removing parts of the bordering vertices from the original polygons, as shown in Figure 1 and Figure 2. A configurable parameter for this component is what is known as the number of points to retain (‘number of PTR’ throughout the rest of the paper), which dictates the percentage of the number of points to keep after discarding the other vertices from the boundaries. This parameter is currently expert-guided and provided to the system as part of the geospatial query. The outcome of this component is a reduced polygon file with a smaller number of boundary vertices.

We then pass this new vector geospatial polygons file to a *geohash encoder* component, which then generates a list of geohashes completely covering the study area (e.g., city), as shown in Figure 5, with Figure 5a showing the covering geohash at precision 6 for the original polygon file of NYC in the USA, while Figure 5b shows the covering geohash for the reduced polygon file that was subjected to the front-stage RDP-based filter previously.

This geohash list (the reduced covering geohash) acts as a quick-and-dirty sieve as follows. For each geospatial point that our system receives from the georeferenced data stream, we generate the corresponding geohash value using a geohash encoder. Our system then applies a front-stage filter subcomponent, which filters and discards points that do not have a matching geohash value in the covering geohash list that our system generated previously. Those are the points that typically fall on the borders and vertices of the original covering geo. The online (dynamic) mode of operation works pretty much in a similar way, considering that we consider a micro-batch processing model, where data streams are aggregated into small packets known as micro-batches before being sent to the processing component. In this sense, the system proceeds in the same way as described for the batch mode, applying the same mechanism to each micro-batch independently. Either way, the remaining points are fed to an online/batch *stratified-like spatial sampler* along the workflow pipeline of our GeoRAP system for further processing.

In more detail, the geohash cover list size is reduced from 1730 to roughly 1471 in the case shown in Figure 5, a reduction that roughly equals 14.9%. The output of this component is a list of geohash values covering the reduced polygons file. This acts as a dictionary to define the stratum of the strata that covers the city area. In this case, each geohash value is a stratum, and all geohash values covering the city are strata. We then pass this cover as a dictionary to our *geohash-based stratified-like geospatial sampler* component. Our sampler works as follows. It groups geospatial tuples in the database (e.g., vehicle mobility data) by their geohash values, and then it selects from each geohash group the same percentage of tuples (e.g., 0.2). In other terms, it selects a percentage of points that corresponds to the sampling fraction from each geohash (equivalent to stratum) independently. It is a stratified-like sampling design that is known to yield better results in terms of accuracy-based QoS constraints. The remaining points are then fed to a *geospatial approximator* downstream in the processing pipeline so that they participate in generating a response to the continuous geospatial query request stepwise. The approximator currently supports two widely spread geospatial queries, which are Top-N and single geo-statistics (such as ‘mean’ and ‘count’). Algorithm 1 shows the mechanism by which GeoRAP operates.
**Algorithm 1:** GeoRAP Workflow/* geohashSize: geohash size, lsa: line simplification algorithm*/Input: geo-stream, SpatialQuery (SQ), samplingRate, polygons, geohashSize, lsa//lsa: Ramer-Douglas-PeuckersimplifiedPolygons = lineSimplification(polygons, lsa)geoCover ← computeGeoCover (simplifiedPolygons, geohashSize)strata = stratify (geoCover)Foreach query window time interval do  repSample = ∅//tuples sampled in current time window  Foreach stratum in strata do     stratumSample = sample (stratum, samplingRate)      repSample.add(stratumSample) End   //Calculate and feed incremental result every time window stepwiseOutput ← execute (SQ, repSample) return stepwiseOutput w/rigorous error bounds (i.e., standard error, distance, correlation coefficient, EMD)End

### 4.3. Spatial Queries Supported

Since we have two modes of operation (batch and online/dynamic), we support the same sets of geospatial queries for both modes. In more detail, GeoRAP currently supports batch processing of two core spatial analytics types. Those are single queries such as ‘count’ and ‘mean’ and aggregation queries such as Top-N. It also supports two kinds of Spatial Continuous Queries (SCQ) over fast-arriving geospatial big data streams. Those are geo-statistical queries and ensembles (Top-N). Both types of queries require online stateful aggregations, which are computationally expensive. By applying line generalization as a frontline filtering stage intrinsically within the layers of GeoRAP, we aim at achieving plausible gains in time-based QoS goals (i.e., lowering response time and latency) while keeping accuracy loss in check and within plausible statistical figures. A geo-statistical query may request “finding the average speed of taxi trips in each administrative area of a city across time”. On the other hand, ensembles include Top-N queries that may request “finding zones with most taxi mobility densities across time”.

An integral part of GeoRAP depends on applying a representative sampling method that is based on the theory of stratification [10]. Hence, we first compute the estimated average of the study target variable using ${\overset{-}{Y}}_{\mathrm{GeoRAP}}={\hat{t}}_{\mathrm{GeoRAP}}/N=\sum_{k=1\;}^{K}(N_{k}/N){\overset{-}{y}}_{k}$, where ${\hat{t}}_{\mathrm{GeoRAP}}=\sum_{j=1}^{J}t_{j}=\sum_{j=1}^{J}N_{j}{\overset{-}{y}}_{j}$ is the estimated total using the sample.

### 4.4. Error Bounds Calculation

Since we have two modes of operation where we support the same sets of queries, our plan for evaluation encloses two modes: online and batch. Having said that, in this section, we discuss the two sets of tools that we employ to test the various functionalities of GeoRAP in both modes of operation, batch and online.

#### 4.4.1. Batch Mode Error Bounds Calculation

For Top-N queries, a kind of stateful aggregation, we rely on a retrofitted and adapted version of the Spearman Correlation Coefficient (SCC) [47] to measure the system’s accuracy-based performance. We use this information-theory tool to measure the accuracy in terms of the system’s ability to retain the aggregation’s original ranking. SCC is a measure of statistical dependency between the ranking of two variables. We have retrofitted it as follows: we take the ranking that results from each method (our GeoRAP method against the baseline), then we serve those to the plain Spearman’s rho, and we apply (1).
(1)$$\rho_{\mathrm{rg}}=\mathrm{cov}(\mathrm{ran}k_{\mathrm{GeoRAP}},\;\mathrm{ran}k_{\mathrm{baseline}})\;/\;(\sigma_{\mathrm{ran}k_{\mathrm{GeoRAP}}}.\sigma_{\mathrm{ran}k_{\mathrm{baseline}}})$$
where $\rho_{\mathrm{rg}}$ is the Spearman’s correlation coefficient applied for ranking statistics, $\mathrm{cov}(\mathrm{ran}k_{\mathrm{GeoRAP}},\;\mathrm{ran}k_{\mathrm{baseline}})$ is the covariance of the rank variables, $\sigma_{rank_{GeoRAP}}\;\mathrm{and}\;\sigma_{rank_{baseline}}$ are the standard deviations of the rank variables for GeoRAP and baseline, respectively.

We also use a distance measurement, specifically Jensen–Shannon divergence (JSD), to measure the distances between the probability distributions of the new method and the baseline. We apply JSD to measure the accuracy of the system for single geo-statistics queries such as “count”. Given two distributions, P and Q, the JSD is defined as in (2).
(2)$$JSD(P|\left|Q\right)=\frac{1}{2}\;KLD(P|\left|R\right)+\frac{1}{2}\;KLD(Q|\left|R\right).$$
where KLD is the Kullback–Leibler divergence, and $R=\frac{1}{2}\;(P+Q)$ is the mixture distribution of P and Q.

#### 4.4.2. Online Mode Error Bounds Calculation

Online approximate query processing is naturally tied to a degree of uncertainty, which needs to be accurately quantified to validate the efficiency of the newly introduced system. Having said that, we apply $SE({\bar{y}}_{GeoRAP})=\sqrt{\hat{V}({\bar{y}}_{GeoRAP})}$ to calculate the standard error introduced by applying GeoRAP. We use the same measure for the baseline systems to compare. $SE({\bar{y}}_{GeoRAP})$ is the standard error that results from estimating the target variable by depending on the sample instead of the population. $\hat{v}\left({\overset{-}{y}}_{GeoRAP}\right)$ is the estimated variance of the estimated average.

### 4.5. Some Primary Implementation Insights

We have designed and developed two different prototypes for our proposed GeoRAP solution. The first prototype operates in batch mode, where all input data are persisted in disk drives. The second prototype considers a setting where the mobility data are poured as data streams and ingested by ingestion systems such as Apache Kafka. We have implemented the first prototype in Python, specifically Geopandas and geospatial support libraries in Python, while we implemented the streaming counterpart over Spark Structured Streaming [5] by exploiting the Spark modular architecture. In fact, by considering the second prototype, GeoRAP operates as a geospatial shape-aware quick-and-approximate filter as a front stage ahead of the underlying structured stream processing engine of Spark. It takes advantage of the micro-batch processing model of the underlying codebase by performing sampling micro-batch-wise. In other words, we sample fair amounts from each stratum (geohash in this case) every micro-batch interval. Figure 6 depicts the mechanism by which we intrinsically incorporated GeoRAP within Spark Structured Streaming.

## 5. Experimental Evaluation Work and Performance Results

### 5.1. Deployment Settings and Benchmarking

***Dataset***. For benchmarking and testing the performance of our GeoRAP system functionality, we have employed three georeferenced mobility big datasets, which are publicly available for maximum result reproducibility.

The first dataset is a publicly available Uber pickup dataset from the city of San Francisco in the USA. It is the anonymized GPS coordinates (longitudes/latitudes) of Uber trips forming circa one million and 85k tuples. The second dataset comes from New York City taxicab trip datasets (https://www1.nyc.gov/site/tlc/about/tlc-trip-recorddata.page, accessed on 10 January 2023), consisting of around 1,400,000 tuples, representing data taxi rides for the first month of 2016. We selected the green taxi trip records, which included fields such as GPS locations and itinerary distances. The third dataset consists of 1,155,654 tuples, representing Electric Taxi GPS mobility trips for a day in the Chinese city of Shenzhen [48]. The target variable in these data is ‘speed’. We aim to calculate the average speed.

The first two datasets (NYC taxicab and San Francesco Uber data) are used to test the functionalities of GeoRAP in batch mode, while the third dataset (Shenzhen electric taxi) is employed to test the potential of GeoRAP to run on distributed computing environments (online mode of operation).

We also employ our method on a unique geotagged air quality dataset collected using low-cost air-quality sensors, consisting of circa 634k records, capturing granular air pollution levels such as PM10 and PM2.5, in addition to other greenhouse gases and vehicle’s harmful combustion engines emissions [49].

***Deployment, experimental*** settings, and operating environment. Since we have two prototypes, we have two deployments. We deployed the first batch mode prototype on a VM that runs on Google Collaboratory with 13 GB RAM and 2 vCPU (2 Intel(R) Xeon(R) CPU @ 2.20 GHz). For the online/dynamic prototype, we deployed Microsoft Azure HDInsight Cluster hosting Apache Spark version 2.2.1. It consists of 6 nodes (2 Head + 4 Worker) with 24 cores (Head (2 × D12 v2) nodes and Worker (4 × D13 v2) nodes). Each head node operates on four cores with 28 GB RAM and 200 GB Local SSD memory, and quantities are double those figures for worker nodes.

**Parameter Configurations.** We herein provide the algorithm parameters. We depend on varying the geohash precision in addition to the sampling rate. We vary geohashes between 5 and 6 and rates between 20% and 80% (with 20% step in-between). Also, we vary the percentage of vertices points in the original polygons to retain (‘number of PTR’ hereafter for short). We choose aggressive 1%, moderate 4%, and smooth 80% PTR.

### 5.2. Performance Testing and Results Discussion

We depend on varying parameters. Those are sampling rate and geohash precision for the baseline method, whereas, in addition, for the new method GeoRAP, we rely on varying a configurable parameter known as the ‘number of PTR’, which is a number between 0 and 100 that dictates how aggressively to simplify the geospatial polygons. Based on that, we vary the ‘number of PTR’ between 1% (permissive), 4% (moderate), and 80% (stringent). We have two testing configurations, one for the batch mode of operation and the other one for the online mode.

#### 5.2.1. Top-N Queries (Batch Mode)

We specifically tested the performance of our prototype against the baseline in terms of time-based and accuracy-based QoS constraints.

We have tested the performance of the following query on two datasets: the Uber San Francesco dataset in addition to the NYC taxicab pickup dataset. The query is “what are the Top-N regions in the city (be that San Francesco or NYC) in terms of taxi pickups (or Uber)”. We vary the sampling fraction as discussed previously and the “number of PTR” in addition to the geohash precision, and then we calculate the Spearman Correlation Coefficient (SCC) for both datasets for the new method of sampling against the baseline (without line simplification). For San Francesco Uber data, we obtain results shown in Figure 7a for geohash precision 6 and “PTR” 1%, while Figure 7b shows the SCC values we obtain for geohash precision 5.

We have noticed that for both configurations (geohash 6 and geohash 5), we obtain statistically significant results for SCC for the GeoRAP against the naïve baseline, with a reduction of the accuracy in SCC for the geohash precision 5 that equals roughly 0.550%, on average, which is statistically insignificant. Figures are similar for geohash precision 6 as we have obtained roughly a 0.554% reduction of the accuracy using SCC. This reveals the fact that the selection of geohash precision has only a very tiny discernible effect on the Top-N loss in the accuracy, where the lower the geohash precision, the more Top-N accuracy we obtain.

We now discuss the results for the NYC data. Figure 8 shows the results of Top-N accuracy using Spearman Correlation Coefficient (SCC) with a ‘number of PTR’ that equals 1% and geohash precision 6 in Figure 8a and geohash precision 5 in Figure 8b. We obtain a reduction in the accuracy that equals roughly 1% for geohash 6 against 1.9% for geohash 5. In this case, the effect of geohash precision on the reduction in the accuracy is more significant than that of the San Francesco data, where granular geohash precision (6 is granular, while 5 is coarser) means more Top-N accuracy.

We have also tested our system with low-cost air quality data coming from NYC. This proves the applicability of our system in various contexts of smart cities where massive amounts of sensor data are captured on a granular scale.

For geospatial Top-N queries, we obtain the results shown in Figure 9. Specifically, Top-N accuracy using Spearman Correlation Coefficient (SCC) with ‘number of PTR’ that equals 4% and geohash precision 6 in Figure 9a, and ‘number of PTR’ that equals 80% in Figure 9b. We obtain a reduction in the accuracy that equals roughly 1.08% for geohash 6 and PTR 4%, against 0.04% for PTR 80%. In this case, the effect of PTR on the reduction in the accuracy is clear, where higher PTR means more Top-N accuracy.

#### 5.2.2. Geo-Stats (Count Queries)—Batch Mode

We have specifically tested the performance of GeoRAP for geo-stats single queries using the following query: “what are the counts of taxi (or Uber) pickups in each region in the city (be that San Francesco or NYC)”.

For this kind of query, we have tested the performance of both systems, the novel GeoRAP and the baseline, using a distance measurement, specifically Jensen–Shannon (JS).

For San Francesco data, we again vary the geohash precision, sampling fraction, and PTR, and we obtain the results shown in Figure 10.

For geohash precision 6 and the number of PTR 1%, the average reduction in accuracy for GeoRAP against the baseline is roughly 17.8%. It was noticed that the reduction in accuracy decreased as we increased the sampling rate to reach roughly 15.6% at a sampling rate that equals 80%. The discernible trend is similar for the same geohash precision at 6, with a greater number of points to retain PTR (4%), as shown in Figure 10b, where we obtain less reduction in accuracy as we increase the sampling rate. However, the performance is better as we obtain an average of 16.2% reduction in accuracy as compared to the 1% PTR counterpart. This conforms to the natural fact that accuracy improves as we increase the number of PTR. This is so because by increasing the number of PTR, we have more data that resemble the source data; thus, we obtain more accurate results.

We obtained similar trends for NYC data, as shown in Figure 11. However, for these data, we obtain a higher reduction in accuracy for the number of PTR 1%, which equals roughly 21.4% for geohash precision 6, decreasing linearly from around 24.9% at a sampling rate of 20 percent to circa 15% at the highest sampling rate of the experiments at 80 percent. For PTR 4% and geohash precision 6, we obtain better accuracy results, as shown in Figure 11b. We obtained a reduction in accuracy that is, on average, 18%, ranging from 22% to 13%. This implies that the system is amenable to a straightforward increase in the sampling rates, meaning that more data results in better accuracy, which is statistically desirable.

Results are even better with a permissive number of PTR (specifically 80% PTR), where we obtain an average reduction in accuracy at roughly 11.3%, which reaches its best at a sampling rate of 80%, where the reduction in accuracy decreases to roughly 8.3%, as shown in Figure 12. The situation is even better for the NYC dataset as we obtain a reduction in accuracy that is roughly 0.36%, on average, ranging from 0.9 at a sampling rate of 0.2 to roughly 0.07 at a permissive sampling rate of 0.8. This means that the higher the sampling rate, the less reduction we obtain in accuracy (desirable).

We measured the JS divergence value on a moderate number of PTR (equals 4%) and a high number of PTR (80%) to statistically shape the relationship between the change in the number of PTR and the JS divergence value as an accuracy measurement. Results obtained conform with the fact that the more the PTR, the better the performance, and the greater the sampling rate for any PTR value, the better the performance in terms of single geo-statistical queries such as ‘count’.

We have also measured the file size in kilobytes for GeoRAP versus the baseline, as shown in Figure 13. As noticed, the file size decreases as we decrease the number of PTR to reach roughly 105 kilobytes (KB) at a number of PTR that equals 1%.

Also, there is a reduction in the number of geohash distinct values covering the remaining embedding area (which is desirable) to reach its lowest, which equals 711 at a PTR that equals 1%. This is significant for performing the kind of spatial join queries that rely on the filter-and-refine approach. It is worth mentioning that the original number of geohash distinct values is 725. Despite being a roughly small difference, it significantly contributes to lowering the running time of Top-N and count queries, as shown in Figure 14.

Figure 14 also shows the remaining tuples after reduction, which is very tiny and explains the insignificant loss in the accuracy discussed previously. On the other hand, we obtain a reduction in the running time at an average of roughly 13.5, while the reduction in the corresponding number of tuples is roughly 2.2, on average, which is insignificant as a trade-off for the significant reduction in the running time. This has an impact on the overall performance of the system for very big geospatial data approximate query processing.

We have also tested our system with low-cost AQ data coming from NYC. This proves the applicability of our system in various contexts of smart cities where massive amounts of sensor data are captured on a granular scale.

For geo-stats, we obtain the results shown in Figure 15. Similar to what we have obtained for sensor mobility data, we obtain a reduction in accuracy on par with 2.029% for a PTR equal to 4 percent. For the PTR that is equal to 80 percent, we obtain a reduction in JS accuracy that is roughly equal to 0.3325. Results obtained from low-cost air quality sensors corroborate the findings from the mobility sensors data in the sense that they assure the validity of our GeoRAP system in providing accurate results with plausible and statistically significant error margins.

#### 5.2.3. Testing Performance of Dynamic Operation Mode

To capture the potentiality of our GeoRAP system as a modular architectural framework that can be ported to distributed computing environments, we tested our system in a distributed environment and measured the performance based on the following geo-statistical continuous spatial query: “find the average speed of a Shenzhen taxicab itinerary trip for one day”.

We vary geohash precisions from 5 to 6 and the sampling rate from 20% to 80% (with a step of 20%). Figure 16 shows the comparison between the plain stratified-like baseline and the new GeoRAP method in the language of accuracy for the estimator of the average value of a target variable (taxi speed in this case).

We notice from Figure 16 that the new method GeoRAP either slightly outperforms or performs on par with the stratified-like baseline. This perfectly meets our target, which is to reduce the response time (i.e., obtaining higher throughput) but at the same time either minimize the error bounds or keep them untouched as compared to the state-of-the-art baseline.

The standard error (SE) increases with a smaller “number of PTR”. So, at 2% “number of points to retain”, we obtain an increase in the standard error that is equal to roughly 4.25%, on average, as compared with the ‘number of PTR’ 5%. This is expected as the percent of 2% is a very aggressive simplification, saving a huge amount of hard disk space, but at the same time increases the standard error slightly. However, this tiny increase in the SE is still statistically acceptable as the algorithm at this point is still on par with the baselines of stratified-like sampling without simplification at geohash precision 6 and stratified-like sampling at the same geohash precision with the coarser level of stratification at the neighborhood level, while outperforming stratified-like sampling with no simplification at geohash precision 5 by a plausible magnitude. In more detail, the data that have been discarded mostly belong to the borders, so they naturally constitute outliers and do not contribute to the target of estimating the average ‘speed’ in heavily trafficked areas of the city. This explains the reason behind obtaining better accuracy (in terms of lowering the SE) by the new method GeoRAP, whereby configuring the “number of PTR” to 5%, we obtain, roughly on average, a 6.1% decrease in the SE by applying GeoRAP instead of the baseline at geohash precision 6.

We also tested our system GeoRAP in distributed systems with NYC taxicab sensor mobility data, and we obtained the results shown in Figure 17. The target variable is the ‘trip distance’, where we seek an answer to a continuous spatial query such as the following: “what is the average trip distance travelled by taxis in NYC in 2016 for each neighbourhood”. This time, we varied the PTR between stringent 4% and permissive 80%. For the PTR 4%, we obtain a reduction in SE that is roughly equal to 0.86%; for PTR 80%, we obtain a reduction in SE that is equal to circa 2.45%, a difference in reduction that is roughly equal to 1.6%, on average. Those numbers agree with the corroboration that we concluded based on the Shenzhen electric taxi dataset.

We have also tested our system in cluster computing mode using NYC low-cost air quality sensor data. We obtained the results shown in Figure 18. The target variable is the ‘particulate matters 2.5, PM2.5′, where we seek an answer to a continuous spatial query such as the following: “what is the average PM2.5 value in each region of NYC city in 2021”. This time, we varied the PTR between stringent 4% and permissive 80%. For the PTR 4%, we obtain a reduction in SE that is roughly equal to 1.76%; for PTR 80%, we obtain a reduction in SE that is equal to circa 2.817%, a difference in reduction that equals roughly 1.07%, on average. Again, those numbers agree with the corroboration that we concluded based on the Shenzhen electric taxi dataset and NYC taxi datasets, as shown in Figure 16 and Figure 17, respectively.

To quantify the gain in the time-based QoS goals, we calculate the throughput as the median (i.e., 50th percentile) of running the same experiments over the same query ten times. Figure 19 shows that GeoRAP outperforms the baseline by roughly 2.6 higher throughput on average. This is attributable to the fact that approximate geo-statistic estimators over geospatial data streams that depend on the theory of stratification involve stateful aggregation, which results in a need to manage state keys stepwise as the system receives data online. Since we are applying a line generalization algorithm at the front stage, we have reduced the number of covering geohash keys, which consequently reduces the number of keys that need to be managed. At 2% “number of PTR”, we reduce the number of keys to manage by roughly 2%. We have noticed that the number of points that fall within the brackets of the discarded geohashes is equal to one point only. This means that we obtain a significant reduction in the number of keys to manage while not compromising the original dataset. This also amplifies the corroboration that only a few points reside on the borders that have been discarded by the simplification algorithm. Those points are considered outliers and do not contribute to the calculation of the geo-statistics.

#### 5.2.4. Testing the Ability to Generate Region-Based Aggregate Geo-Maps

We have tested the ability of GeoRAP to generate region-based aggregate geo-maps, specifically choropleth maps. Those are maps generated in a thematic approach where the study region is divided into administrative divisions (e.g., districts or boroughs in a city), then each division is assigned a specific color selected from a color-coding scheme in such a way that reflects the density of geospatial objects in each division, and typically more dense regions are assigned darker colors.

Figure 20a shows a choropleth map generated from the source original data based on the original geohash covering before applying the RDP algorithm, while Figure 20b shows the choropleth map for the same city with the same source data but with the RDP algorithm applied to the polygon representing the city with a number of PTR that equals 1%. It is obvious that both figures are identical in revealing the density of taxi pickups in NYC for the same period.

It is obvious that both choropleth maps reveal the same density distribution of taxi pickups in NYC. We use visualizations to reveal patterns, which helps us in making decisions regarding city planning; thus, it is unnecessary for the borders to be accurate for the region-based aggregate geo-maps type of visualization. Although the geo-map in Figure 20b is an approximation, it is sufficient for decision making and urban planning. For example, unleashing the points-of-interest (POIs) that are highly dense to decide so as to show whether those areas are highly affected by higher levels of unhealthy pollutants such as particulate matter (PM10 or PM2) as a consequence of the high circulation of vehicles and the substances delivered by their combustion engines.

## 6. Conclusive Remarks

In this paper, we show our design and implementation of a novel system that we term GeoRAP for efficient approximate query processing of disk-resident and fast-arriving georeferenced big data streams. Our solution is based on a filter-sample approach where we first pass the parameters to a front-stage filter to reduce the number of points to accept downstream with a cheap line simplification algorithm (specifically the Douglas–Peuker algorithm). Thereafter, an efficient spatial online sampling method is applied to select a highly representative sample from the remaining points and serve the sample to an online geospatial approximator that computes the query answer (stepwise incrementally for the online mode of operation). GeoRAP has a notable impact on dynamic smart city scenarios that require performing analytics on fast-arriving georeferenced data streams [27,50]. We have compared GeoRAP with a plain counterpart that does not feature polygon simplification in the front stage to reduce the allowed input data. We have tested our system with various sensor georeferenced data streams coming from mobility GPS sensors and low-cost air quality sensors. We specifically tested the system performance on geospatial aggregate (e.g., Top-N) and geo-stats (e.g., ‘average’, ‘count’) queries. We obtained significant speedups and throughput while also reducing the polygon file sizes and keeping the accuracy performance under control. In more detail, our results corroborate the fact that increasing our configurable variable ‘number of PTR’ improves accuracy. In other terms, more ‘PTR’ means better performance, and more sampling rate for any PTR value means better performance in terms of single geo-statistical queries such as ‘count’ and also geospatial aggregation such as Top-N. Those accuracy figures are obtainable while being able to significantly reduce the polygon file size, thus lowering the load on the network traffic and providing significant speedups. We conclude that polygon and line simplification are indispensable for processing, managing, and analyzing huge amounts of sensor’s georeferenced big data with QoS guarantees. This corroborates Tobler’s first law of geography, which dictates that nearby spatial objects are more related than those that are far apart. This means that more than often, in smart environment applications, we consider analyzing data cluttered within city boundaries more often than we do for those on the outskirts and near the administrative boundaries. Having said that, simplifying polygons has a direct impact on time-based QoS goals, such as lowering latency while keeping quality-based QoS goals in check, such as obtaining a higher accuracy of estimates, because ultimately, the DP algorithm forces an AQP instead of deterministic counterparts. Having said that, we advocate applying the DP algorithm for polygon and line simplification while working on analyzing and managing highly skewed and massive amounts of georeferenced sensor data streams such as mobility data and low-cost air quality data.

As a future research frontier, we consider incorporating GeoRAP with a spatial join processor so as to enhance the performance of joining georeferenced data streams with enrichment disk-resident geographical tables, such as in [19]. We also consider designing and implementing a modified version of GeoRAP so that it performs geospatial sampling over georeferenced time-series data (also known as space–time series) [26].

Also, one of the known limitations of DP is that it is a batch algorithm that needs data to persist in disks to operate [28]. A future direction is to design an adaptive online version of the DP algorithm, which can be adaptively modified upon receiving more tuples upstream. Also, we need an online method to profile stream data and account for the fact that we do not discard edges of the polygons abruptly; instead, we depend on well-established statistics, for example, performing online density-based clustering to check the distribution of data before clipping any corresponding edge from the polygons. Our architecture is modular, and various line simplification algorithms are easily pluggable. In a future work, we intend to compare the performance of our system by employing other line simplification algorithms.

## Figures and Tables

**Figure 1 sensors-23-08178-f001:**
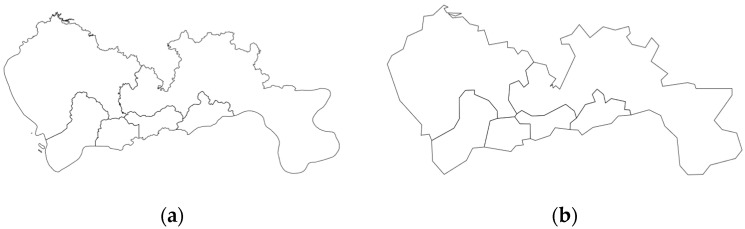
Applying Ramer–Douglas–Peucker line simplification algorithm to the administrative polygons representing Shenzhen city in China by using number of points to retain, which is equal to roughly 2%, which means we retain 2 percent of the total number of points from the original lines that are constituting the borders and the administrative separating lines of the polygons. (**a**) before applying the RDP algorithm; (**b**) after applying the RDP algorithm. For (**b**), the edges of the outskirt lines and the separating lines are sharper than those in (**a**).

**Figure 2 sensors-23-08178-f002:**
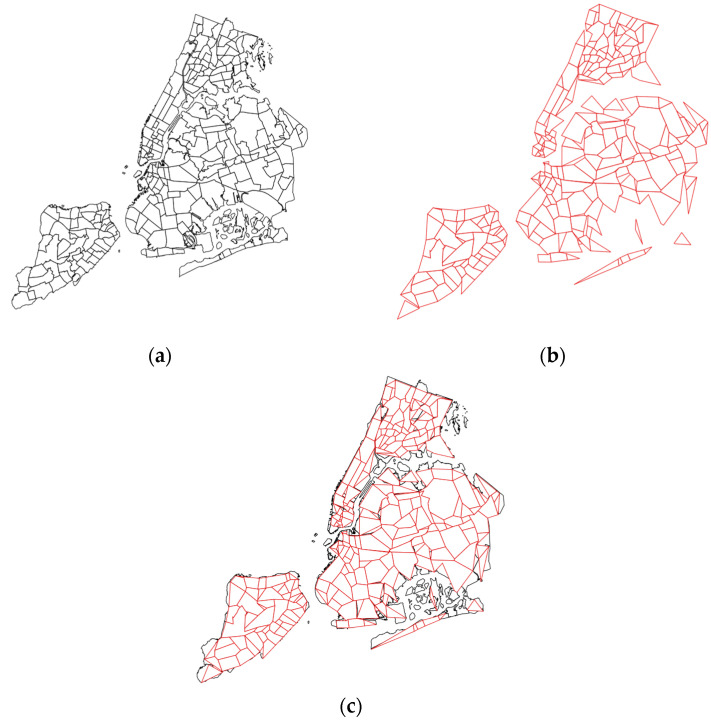
Applying Ramer–Douglas–Peucker line simplification algorithm to the administrative polygons representing NYC city in USA by using number of points to retain, which is equal to roughly 1%, which means we retain 1 percent of the total number of points from the original lines that are constituting the borders and the administrative separating lines of the polygons. This simplification is considered aggressive and may result in elimination of few polygons, as shown in (**c**). (**a**) shows original NYC polygon map, while (**b**) shows reduced NYC polygon map with RDP algorithm applied at number of PTR that is equal to 1%; in (**c**), we show the original NYC polygons map overlayed with reduced NYC polygons map with Ramer–Douglas–Peucker (RDP) applied with number of PTR that is equal to 1%.

**Figure 3 sensors-23-08178-f003:**
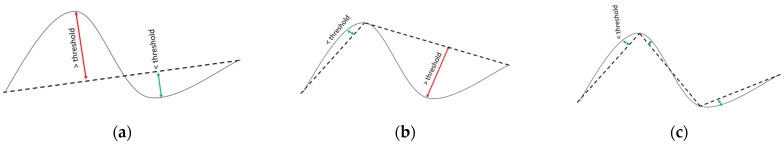
Applying Ramer–Douglas–Peucker line simplification algorithm (**a**) Original line and line segment. (**b**) First step, simplification of the DP algorithm. (**c**) Second step, simplification of the DP algorithm.

**Figure 4 sensors-23-08178-f004:**
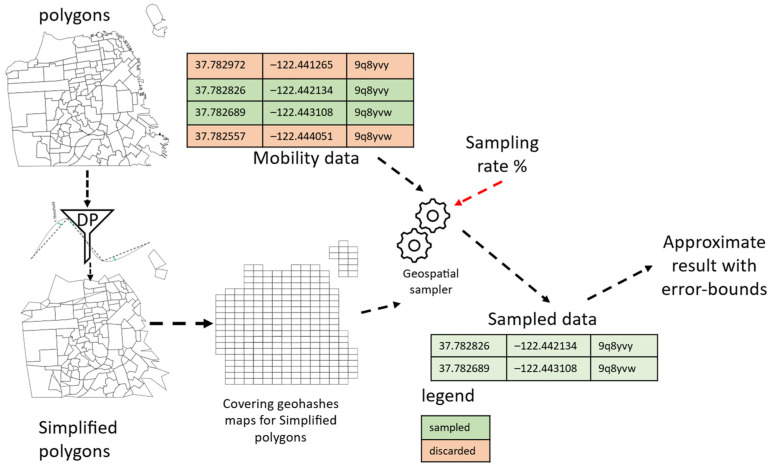
GeoRAP architecture.

**Figure 5 sensors-23-08178-f005:**
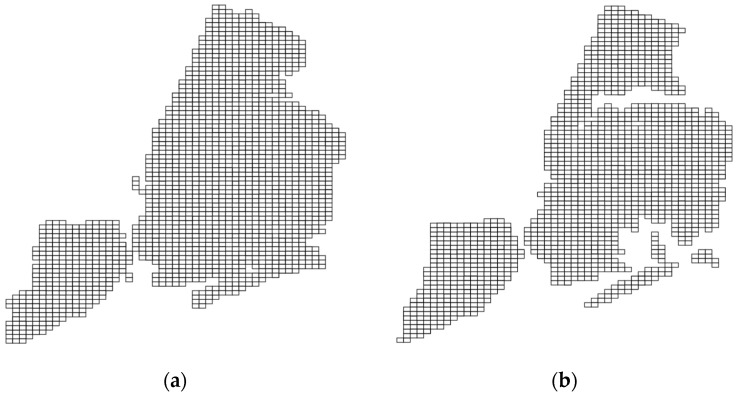
Geohash cover at precision 6, generated to the reduced polygon for the administrative polygons representing NYC in USA, which has been subjected to RDP algorithm with “number of PTR” that equals 1%. (**a**) geohash cover of the original polygons; (**b**) geohash cover for the reduced polygons subjected to RDP with PTR 1%.

**Figure 6 sensors-23-08178-f006:**
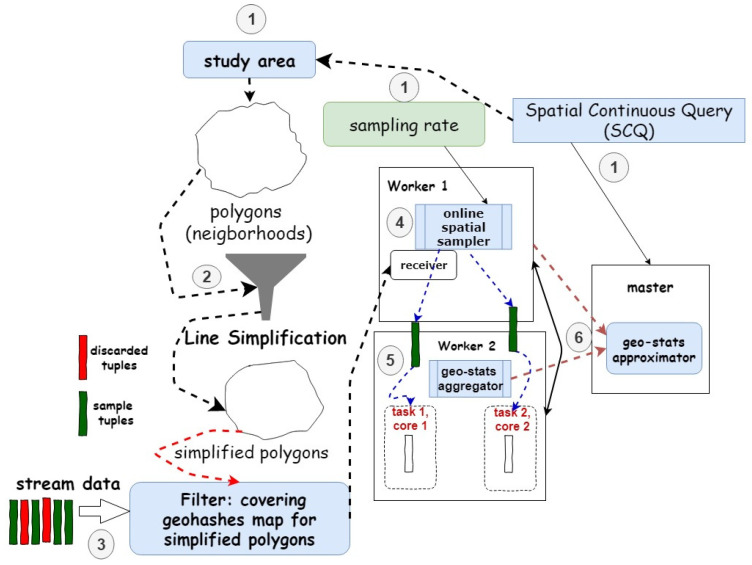
GeoRAP implementation using Apache Spark Structured Streaming. Numbers 1 to 6 show the sequence of system operation in online mode above Apache Spark.

**Figure 7 sensors-23-08178-f007:**
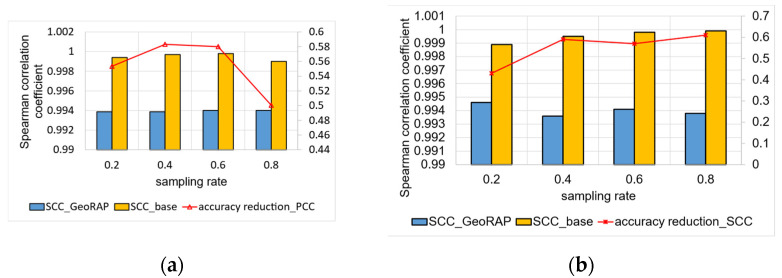
Correlation coefficient (Spearman Correlation Coefficient, SCC) GeoRAP against plain baseline (San Francesco Uber pickup dataset). (**a**) at geohash precision 6 and PTR 1%; (**b**) at geohash precision 5 and PTR 1%.

**Figure 8 sensors-23-08178-f008:**
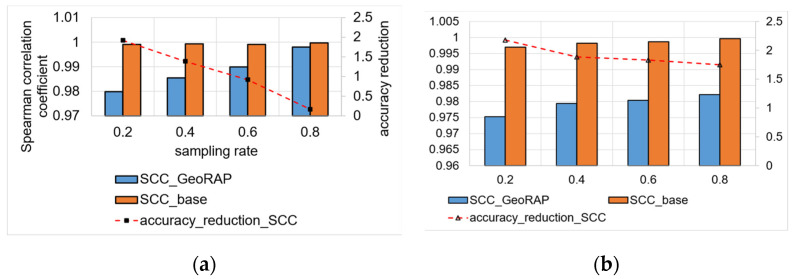
Correlation coefficient (Spearman Correlation Coefficient, SCC) GeoRAP against plain baseline, NYC taxicab dataset. (**a**) at geohash precision 6 and PTR 1%; (**b**) at geohash precision 5 and PTR 1%.

**Figure 9 sensors-23-08178-f009:**
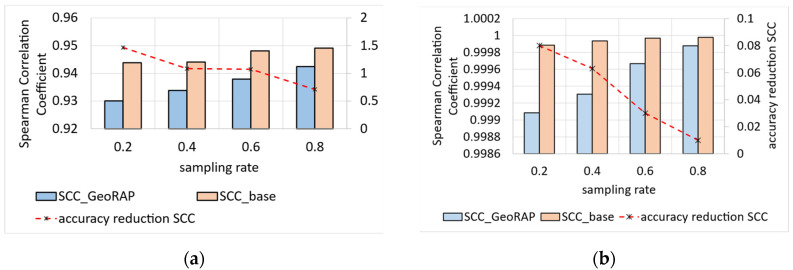
Correlation coefficient (Spearman Correlation Coefficient, SCC) GeoRAP against plain baseline, NYC low-cost sensors air quality dataset. (**a**) at geohash precision 6 and PTR 4%; (**b**) at geohash precision 6 and PTR 80%.

**Figure 10 sensors-23-08178-f010:**
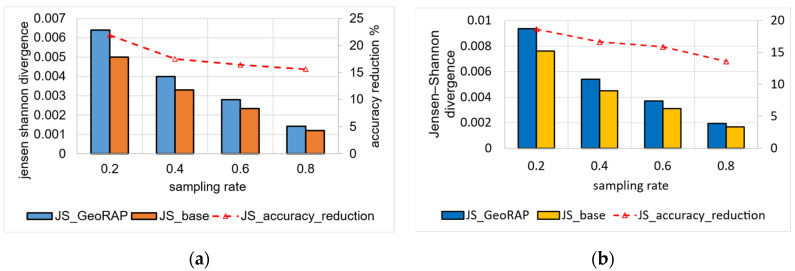
Jensen–Shannon (JS) divergence GeoRAP against plain baseline, San Francesco Uber data. (**a**) at geohash precision 6 and PTR 1%; (**b**) at geohash precision 6 and PTR 4%.

**Figure 11 sensors-23-08178-f011:**
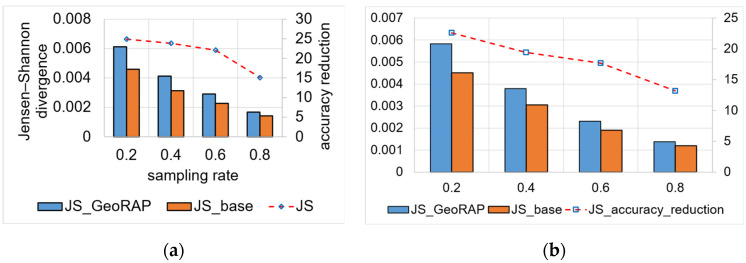
Jensen–Shannon (JS) divergence GeoRAP against plain baseline, NYC data. (**a**) at geohash precision 6 and PTR 1%; (**b**) at geohash precision 6 and PTR 4%.

**Figure 12 sensors-23-08178-f012:**
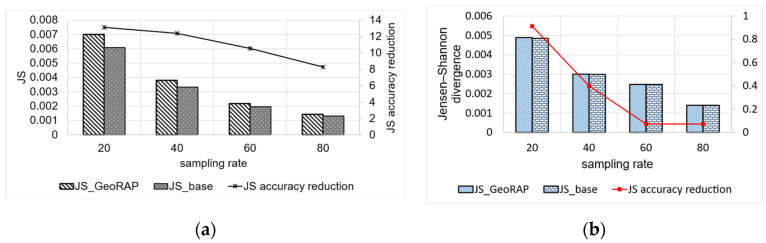
Jensen–Shannon (JS) divergence GeoRAP against plain baseline at geohash precision 6 and permissive PTR 80%. (**a**) San Francesco Uber dataset; (**b**) NYC taxi pickup datasets.

**Figure 13 sensors-23-08178-f013:**
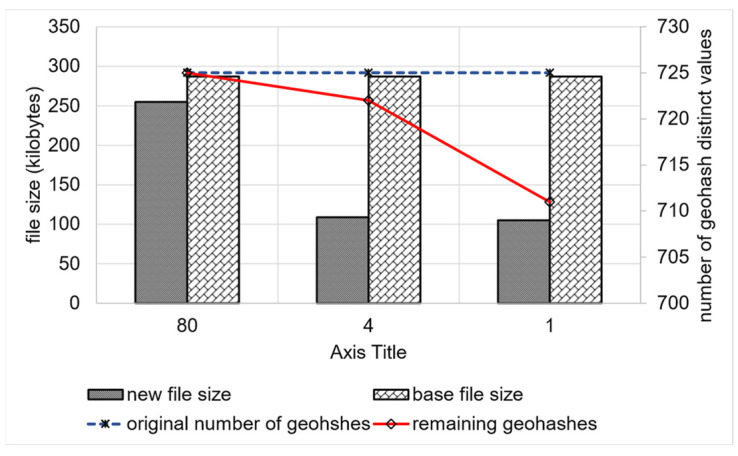
File size GeoRAP against plain baseline at geohash precision 6 and various PTR values, San Francesco data.

**Figure 14 sensors-23-08178-f014:**
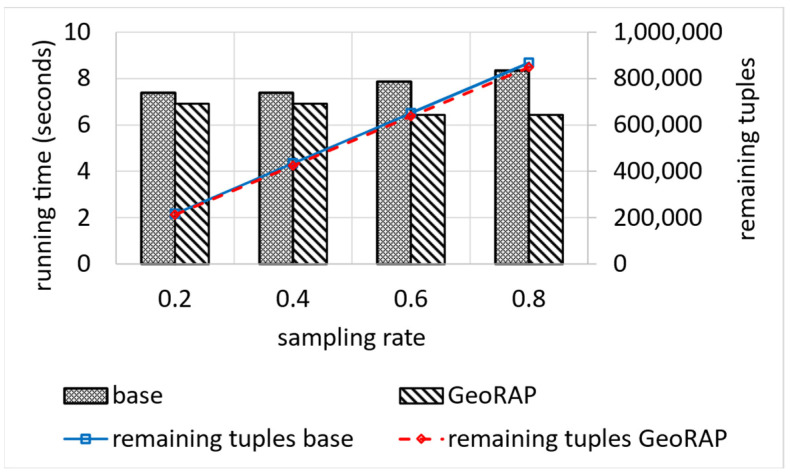
End–end running time GeoRAP against plain baseline at geohash precision 6 and PTR 1%, San Francesco data.

**Figure 15 sensors-23-08178-f015:**
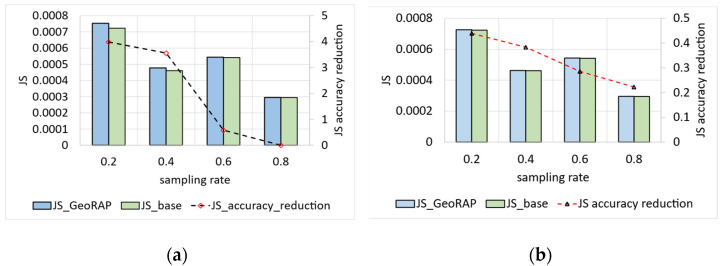
Jensen–Shannon (JS) divergence GeoRAP against plain baseline, NYC low-cost air quality data. (**a**) at geohash precision 6 and PTR 4%; (**b**) at geohash precision 6 and PTR 80%.

**Figure 16 sensors-23-08178-f016:**
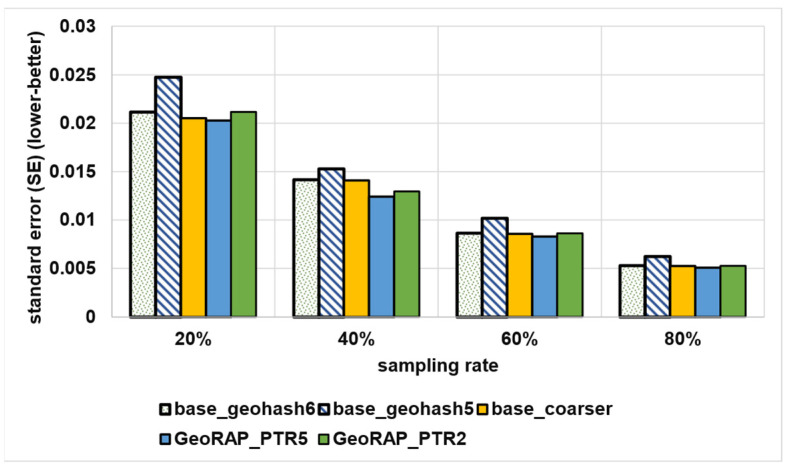
Estimation errors by GeoRAP vs. baseline counterparts.

**Figure 17 sensors-23-08178-f017:**
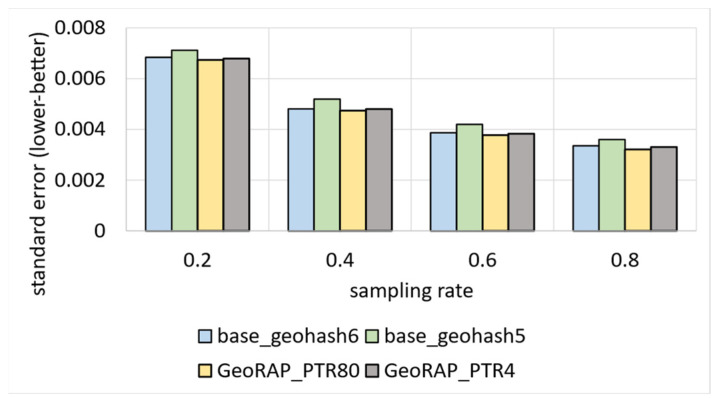
Estimation errors by GeoRAP vs. baseline counterparts, NYC taxicab sensor mobility data.

**Figure 18 sensors-23-08178-f018:**
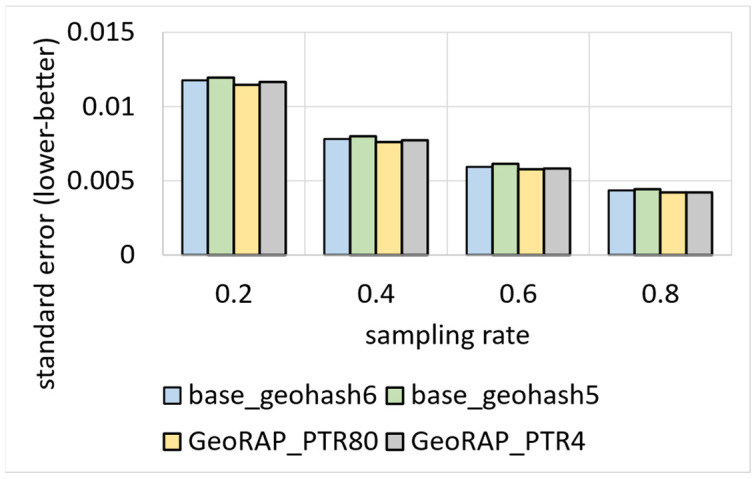
Estimation errors by GeoRAP vs. baseline counterparts, NYC air quality sensor data.

**Figure 19 sensors-23-08178-f019:**
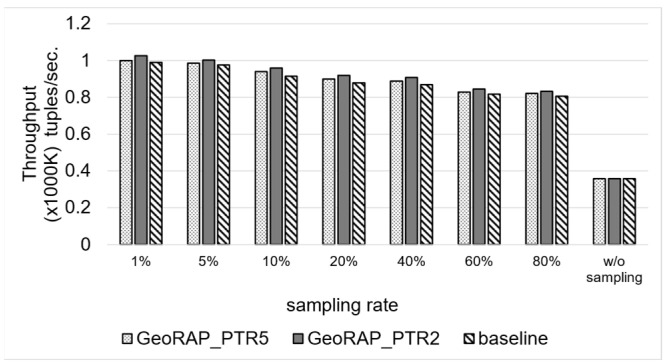
Throughput of GeoRAP against baselines.

**Figure 20 sensors-23-08178-f020:**
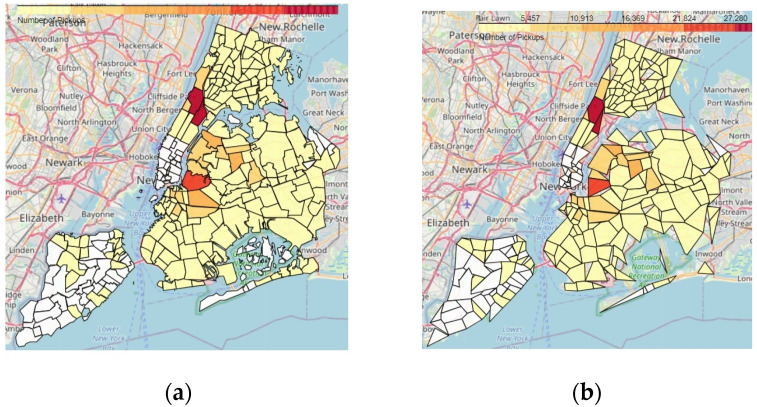
Choropleth map for taxi pickup data NYC USA, GeoRAP against plain baseline at geohash precision 6 and permissive PTR 1%. (**a**) original polygons; (**b**) reduced polygons applying RDP algorithm. For both figures, the darker the color the higher the density in terms of number of spatial objects within a polygon.

## Data Availability

Publicly available datasets were analyzed in this study. NYC taxicabs dataset can be found at https://www1.nyc.gov/site/tlc/about/tlc-trip-record-data.page, accessed on 10 January 2023. Uber San Francesco dataset is available publicly online and can be found at https://raw.githubusercontent.com/dima42/uber-gps-analysis/master/gpsdata/all.tsv, accessed on 10 January 2023.

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
