# Peer review of "Polygon Simplification for the Efficient Approximate Analytics of Georeferenced Big Data"

_sensors, 2023, doi:10.3390/s23198178_

Round 1

Reviewer 1 Report

The paper is well written with a detailed discussion of each section; however, the related works or state-of-the-art methods are missing. Either the introduction section can be extended to cover such discussion, or a new section can be included. Are the results comparable with other methods? This will show the validity of the results generated.

Reviewer 2 Report

The authors need to highlight the motivations why this work is important in the light of relevant literature then the contribution of the work can be appreciated. The authors may need to evaluate the developed approach in an appropriate application relevant to sensors domain thus the work might be more relevant to theme of the journal. In its current form, the work has less relevance to the journal’s domain.

The authors need to highlight the motivations why this work is important in the light of relevant literature then the contribution of the work can be appreciated. The authors may need to evaluate the developed approach in an appropriate application relevant to sensors domain thus the work might be more relevant to theme of the journal. In its current form, the work has less relevance to the journal’s domain.

Reviewer 3 Report

(1) The abstract is the key to the full text, it is too long, and a more concise expression is necessary. In addition, the summary of innovation points is not enough, and the abstract should be re-summarized.

(2) The chapter arrangement of the manuscript should be included in the introduction, and the comparison and summary of the advantages and disadvantages of the various methods are far from sufficient in the introduction.

(3) The literature review given is not highly relevant to the methods proposed by the authors, and the authors should explain the current status of research related to their own research, and the problems and shortcomings that are not solved should be better elaborated.

(4) The typography needs to be perfected, and the manuscript should not have large areas of white space.

(5) The description of the innovation points in Figure 1/2 is too brief, and we want the innovation points to be clearly explained in the diagram.

(6) Simply providing experimental results is not enough. The operating environment of the algorithm in the experiment needs to be explained in detail, and the algorithm parameters need to be provided in detail to replicate the algorithm.

(7) Limitation - There is no information about the limitations of the proposed method, they should include it. And future research direction on how to fix that challenge.

(8) The summary of innovative points in the conclusion should be more specific and quantifiable, and the specific conclusions obtained by the author need to be clearly explained.

(9) The references need to be carefully revised, and key information such as page numbers is not standardized.

Reviewer 4 Report

Aiming at the low efficiency of traditional geo-referenced data stream processing system, this paper designs and implements an online geospatial simplified approximation processing system named GeoRAP. The system uses a sampling approach based on the Ramer-Douglas-Peucker line simplification algorithm and similar spatial hierarchies to simplify geospatial big data in order to efficiently approximate query processing for disk-resident and fast arriving geo-referenced big data streams. By comparing the sampling results of the conventional baseline system and the GeoRAP system, the experiments show that the query efficiency of GeoRAP is significantly better than that of the conventional baseline sampling method and has better accuracy. The paper is clear, original and innovative in its approach, but also suffers from some formatting and content problems:

1)      Table 1 content does not comply with the table specification, it is recommended to modify according to the academic paper table format.

2)      Equation (1) is sandwiched in the middle of the text; it is recommended that the equation be listed separately and cited in the text by serial number.

3)      In terms of experimental evaluation, only one dataset of Shenzhen electric cabs is selected to verify the online dynamic processing capability of GeoRAP system, and the experimental result lacks representativeness, so it is recommended to select more than two datasets for comparative verification.

 There are several grammatical mistakes. Therefore, some editing for English language is required throughout the manuscript.

Round 2

Reviewer 2 Report

The authors have met the concerned raised

No comment

Reviewer 3 Report

This article is  addressed reviewers comments well.